# LiteEval: A Coarse-to-Fine Framework for Resource Efficient Video Recognition

**Zuxuan Wu**[1*]**, Caiming Xiong**[2]**, Yu-Gang Jiang**[3]**, Larry S. Davis**[1]
[1] University of Maryland, [2] Salesforce Research, [3] Fudan University

## Abstract

This paper presents LiteEval, a simple yet effective coarse-to-fine framework for resource efficient video recognition, suitable for both online and offline scenarios. Exploiting decent yet computationally efficient features derived at a coarse scale with a lightweight CNN model, LiteEval dynamically decides on-the-fly whether to compute more powerful features for incoming video frames at a finer scale to obtain more details. This is achieved by a coarse LSTM and a fine LSTM operating cooperatively, as well as a conditional gating module to learn when to allocate more computation. Extensive experiments are conducted on two large-scale video benchmarks, FCVID and ActivityNet, and the results demonstrate LiteEval requires substantially less computation while offering excellent classification accuracy for both online and offline predictions.

## 1 Introduction

Convolutional neural networks (CNNs) have demonstrated stunning progress in several computer vision tasks like image classification [11, 39, 14], object detection [28, 10], video classification [34, 33], *etc*, sometimes even surpassing human-level performance [11] when recognizing fine-grained categories. The astounding performance of CNN models, while making them appealing for deployment in many practical applications such as autonomous vehicles, navigation robots and image recognition services, results from complicated model design, which in turn limits their use in real-world scenarios that are often resource-constrained. To remedy this, extensive studies have been conducted to compress neural networks [2, 26, 20] and design compact architectures suitable for mobile devices [13, 16]. However, they produce one-size-fits-all models that require the same amount of computation for all samples.

Although computationally efficient models usually exhibit good accuracy when recognizing the majority of samples, computationally expensive models, if not ensembles of models, are needed to additionally recognize corner cases that lie in the tail of the data distribution, offering top-notch performance on standard benchmarks like ImageNet [3] and COCO [21]. In addition to network design, the computational cost of CNNs is directly affected by input resolution—74% of computation can be saved (measured by floating point operations) when evaluating a ResNet-101 model on images with half of the original resolution, while still offering reasonable accuracy. Motivated by these observations, a natural question arises: can we have a network with components of different complexity operating on different scales and derive policies conditioned on inputs to switch among these components to save computation? Intuitively, during inference, lightweight modules are run by default to recognize easy samples (*e.g.*, images with canonical views) with coarse scale inputs and high-precision components will be activated to further obtain finer details to recognize hard samples

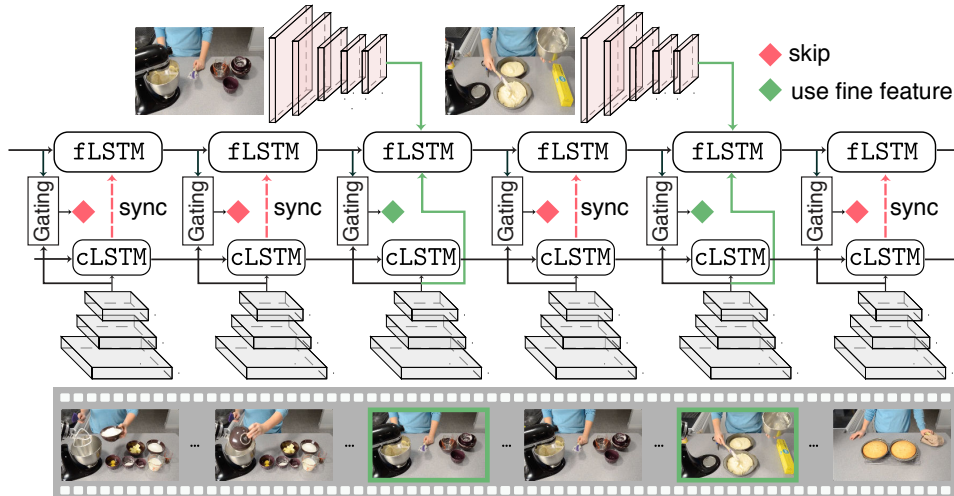

Figure 1: **An overview of the proposed framework.** At each time step, coarse features, computed with a lightweight CNN, together with historical information are used to determine whether to examine the current frame more carefully. If further inspection is needed, fine features are derived to update the fine LSTM; otherwise the two LSTMs are synchronized. See texts for more details.

(*e.g.*, images with occlusion). This is conceptually similar to human perception systems where we pay more attention to complicated scenes while a glance would suffice for most objects.

In this spirit, we explore the problem of dynamically allocating computational resources for video recognition. We consider resource-constrained video recognition for two reasons: (1) Videos are more computationally demanding compared to images. Thus, video recognition systems should be resource efficient, since computation is a direct indicator of energy consumption, which should be minimized to be cost-effective and eco-friendly; additionally, power assumption directly affects battery life of embedded systems. (2) Videos exhibit large variations in computation required to be correctly labeled. For instance, for videos that depict static scenes (*e.g.*, "river" or "desert") or centered objects (*e.g.*, "gorilla" or "panda"), viewing a single frame already gives high confidence, while one needs to see more frames in order to distinguish "making latte" from "making cappuccino". Further, frames needed to predict the label of a video clip not only differ among different classes but also within the same category. For example, for many sports actions like "running" and "playing football", professionally recorded videos with less camera motion are more easily recognized compared to user-generated videos using hand-held devices or wearable cameras.

We introduce LITEEVAL, a resource-efficient framework suitable for both online and offline video classification, which adaptively assigns computational resources to incoming video frames. In particular, LITEEVAL is a coarse-to-fine framework that uses coarse information for economical evaluation while only requiring fine clues when necessary. It consists of a coarse LSTM operating on features extracted from downsampled video frames using a lightweight CNN, a fine LSTM whose inputs are features from images of a finer scale using a more powerful CNN, as well as a gating module to dynamically decide the granularity of features to use. Given a stream of video frames, at each time step, LITEEVAL computes coarse features from the current frame and updates the coarse LSTM to accumulate information over time. Then, conditioned on the coarse features and historical information, the gating module determines whether to further compute fine features to obtain more details from the current frame. If further analysis is needed, fine features are computed and input into the fine LSTM for temporal modeling; otherwise hidden states from the coarse LSTM are synchronized with those of the fine LSTM such that the fine LSTM contains all information seen so far to be readily used for prediction. Finally, LITEEVAL proceeds to the next frame. Such a recurrent and efficient way of processing video frames allows LITEEVAL to be used in both online and offline scenarios. See Figure 1 for an overview of the framework.

We conduct extensive experiments on two large-scale video datasets for generic video classification (FCVID [18]) and activity recognition (ACTIVITYNET [12]) under both online and offline settings. For offline predictions, we demonstrate that LITEEVAL achieves accuracies that are on par with the

strong and popular uniform sampling strategy while requiring 51.8% and 51.3% less computation, and it also outperforms efficient video recognition approaches in recent literature [41, 4]. We also show that LITEEVAL can be effectively used for online video predictions to accommodate different computational budgets. Furthermore, qualitative results suggest the learned fine feature usage policies not only correspond to the difficulty to make predictions (*i.e.*, easier samples require fewer fine features) but also can reflect salient parts in videos when recognizing a class of interest.

## 2 Approach

LITEEVAL consists of a coarse LSTM and a fine LSTM that are organized hierarchically taking in visual information at different granularities, as well as a conditional gating module governing the switching between different feature scales. In particular, given a stream of video frames, the goal of LITEEVAL is to learn a policy that determines at each time step whether to examine the incoming video frame carefully with discriminative yet computationally expensive features, conditioned on a quick glance of the frame with economical features computed at a coarse scale and historical information. LITEEVAL operates on coarse information by default and is expected to take in fine details infrequently, reducing overall computational cost while maintaining recognition accuracy. In the following, we introduce each component in our framework in detail, and present the optimization of the model.

### 2.1 A Coarse-to-Fine Framework

**Coarse LSTM.**   Operating on features computed at a coarse image scale using a lightweight CNN model (see Sec. 3.1 for details), the coarse LSTM quickly glimpses over video frames to get an overview of the current inputs in a computationally efficient manner. More formally, at the $t$-th time step, the coarse LSTM takes in the coarse features $\boldsymbol{v}_t^c$ of the current frame, previous hidden states $\boldsymbol{h}_{t-1}^c$ and cell outputs $\boldsymbol{c}_{t-1}^c$ to compute the current hidden states $\boldsymbol{h}_t^c$ and cell states $\boldsymbol{c}_t^c$:

$$\boldsymbol{h}_t^c,\ \boldsymbol{c}_t^c = \texttt{cLSTM}(\boldsymbol{v}_t^c,\ \boldsymbol{h}_{t-1}^c,\ \boldsymbol{c}_{t-1}^c). \tag{1}$$

**Conditional gating module.**   The coarse LSTM skims video frames efficiently without allocating too much computation; however, fast processing with coarse features will inevitably overlook important details needed to differentiate subtle actions/events (*e.g.*, it is much easier to separate "drinking coffee" from "drinking beer" with larger video frames). Therefore, LITEEVAL incorporates a conditional gating module to decide whether to examine the incoming video frame more carefully to obtain finer details. The gating module is a one-layer MLP that outputs the probability (unnormalized) to compute fine features with a more powerful CNN:

$$\boldsymbol{b}_t \in \mathbb{R}^2 = \boldsymbol{W}_g^\top [\boldsymbol{v}_t^c, \boldsymbol{h}_{t-1}^f, \boldsymbol{c}_{t-1}^f], \tag{2}$$

where $\boldsymbol{W}_g$ are the weights for the conditional gate, $\boldsymbol{h}_{t-1}^f$ and $\boldsymbol{c}_{t-1}^f$ are the hidden and cell states of the fine LSTM (discussed below) from the previous time step, and $[\,,\,]$ denotes the concatenation of features. Since the gating module aims to make a discrete decision whether to compute features at a finer scale based on $\boldsymbol{b}_t$, a straightforward way is choose a higher value in $\boldsymbol{b}_t$, which, however, is not differentiable. Instead, we define a random variable $B_t$ to make the decision through sampling from $\boldsymbol{b}_t$. Learning such a parameterized gating function by sampling can be achieved in different ways, as will be discussed below in Section 2.2.

**Fine LSTM.**   If the gating module selects to pay more attention to the current frame (*i.e.*, $B_t = 1$), features at a finer scale will be computed with a computationally intensive CNN, and will be sent to the fine LSTM for temporal modeling. In particular, the fine LSTM takes as inputs—fine features $\boldsymbol{v}_t^f$ concatenated with coarse features $\boldsymbol{v}_t^c$, previous hidden states $\boldsymbol{h}_{t-1}^f$ and cell states $\boldsymbol{c}_{t-1}^f$—to produce hidden states $\boldsymbol{h}_t^f$ and cells outputs $\boldsymbol{c}_t^f$ of the current time step:

$$\widetilde{\boldsymbol{h}_t^f},\ \widetilde{\boldsymbol{c}_t^f} = \texttt{fLSTM}([\boldsymbol{v}_t^c, \boldsymbol{v}_t^f],\ \boldsymbol{h}_{t-1}^f,\ \boldsymbol{c}_{t-1}^f) \tag{3}$$

$$\boldsymbol{h}_t^f = (1 - B_t)\boldsymbol{h}_{t-1}^f + B_t\widetilde{\boldsymbol{h}_t^f}, \quad \boldsymbol{c}_t^f = (1 - B_t)\boldsymbol{c}_{t-1}^f + B_t\widetilde{\boldsymbol{h}_t^f}. \tag{4}$$

When the gating module opts out of the computation of fine features (*i.e.*, $B_t = 0$), hidden states from the previous time step are reused.

**Synchronizing the `cLSTM` with the `fLSTM`.** It worth noting that the coarse LSTM contains information from all frames seen so far, while hidden states in the fine LSTM only consist of accumulated knowledge from frames selected by the gating module. While fine-grained details are stored in `fLSTM`, `cLSTM` provides context information from the remaining frames that might be beneficial for recognition. To obtain improved performance, a straightforward way is to concatenate their hidden states before classification, yet they are asynchronous (the coarse LSTM is always ahead of the fine LSTM, seeing more frames), making it difficult to know when to perform fusion. Therefore, we synchronize these two LSTMs by simply copying. In particular, at the $t$-th step, if the gating module decides not to compute fine features (*i.e.*, $B_t = 0$ in Equation 4), instead of using $\boldsymbol{h}_{t-1}^f$ directly, we update $\boldsymbol{h}_t^f = [\boldsymbol{h}_t^c, \boldsymbol{h}_{t-1}(D^c + 1 : D^f)]$, where $D^c$ and $D^f$ denote the dimension of $\boldsymbol{h}^c$ and $\boldsymbol{h}^f$, respectively. Similar modifications are performed to $\boldsymbol{c}_t^f$. Now the hidden states in the fine LSTM contains all information seen so far and can be readily used to derive predictions at any time: $\boldsymbol{p}_t = \texttt{softmax}(\boldsymbol{W}_p^\top \boldsymbol{h}_t^f)$, where $\boldsymbol{W}_p$ denotes the weights for the classifier.

## 2.2 Optimization

Let $\Theta = \{\Theta_{\texttt{cLSTM}}, \Theta_{\texttt{fLSTM}}, \Theta_g\}$ denote the trainable parameters in the framework, where $\Theta_{\texttt{cLSTM}}$ and $\Theta_{\texttt{fLSTM}}$ represent the parameters in the coarse and fine LSTMs, respectively and $\Theta_g$ are weights for the gating module [1]. During training, we use predictions from the last time step $T$ as the video-level predictions, and optimize the following loss function:

$$\underset{\Theta}{\text{minimize}} \ \mathbb{E}_{\substack{B_t \sim \texttt{Bernoulli}(\boldsymbol{b}_t; \Theta_g) \\ (\boldsymbol{x}, \boldsymbol{y}) \sim D_{train}}} \left[ -\boldsymbol{y} \log(\boldsymbol{p}_T(\boldsymbol{x}; \Theta)) + \lambda(\frac{1}{T} \sum_{t=1}^T B_t - \gamma)^2 \right]. \tag{5}$$

Here $\boldsymbol{x}$ and $\boldsymbol{y}$ denote a sampled video and its corresponding one-hot label vector from the training set $D_{train}$ and the first term is a standard cross-entropy loss. The second term limits the usage of fine features to a predefined target $\gamma$ with $\frac{1}{T} \sum_{t=1}^T B_t$ being the fraction of the number of times fine features are used over the entire time horizon. In addition, $\lambda$ balances the trade-off between recognition accuracy and computational cost.

However, optimizing Equation 5 is not trivial as the decision whether to compute fine features is binary and requires sampling from a Bernoulli distribution parameterized by $\Theta_g$. One way to solve this is to convert the optimization in Equation 5 to a reinforcement learning problem and then derive the optimal parameters of the gating module with policy gradient methods [29] by associating each action taken with a reward. However, training with policy gradient requires techniques to reduce variance during training as well as carefully selected reward functions. Instead, we use a Gumbel-Max trick to make the framework fully differentiable. More specifically, given a discrete categorical variable $\hat{B}$ with class probabilities $P(\hat{B} = k) \propto b_k$, where $b_k \in (0, \infty)$ and $k \leq K$ ($K$ denotes the total number of classes; in our framework $K = 2$), the Gumbel-Max [9, 23] trick indicates the sampling from a categorical distribution can be performed in the following way:

$$\hat{B} = \underset{k}{\arg \max}(\log b_k + G_k), \tag{6}$$

where $G_k = -\log(-\log(U_k))$ denotes the Gumbel noise and $U_k$ are i.i.d samples drawn from $\texttt{Uniform}(0, 1)$. Although the $\arg \max$ operation in Equation 6 is not differentiable, we can use `softmax` as as a continuous relaxation of $\arg \max$ [23, 17]:

$$B_i = \frac{\exp((\log b_i + G_i)/\tau)}{\sum_{j=1}^K \exp((\log b_j + G_j)/\tau)} \quad \text{for } i = 1, .., K \tag{7}$$

where $\tau$ is a temperature parameter controlling discreteness in the output vector $B$. Consider the extreme case when $\tau \to 0$, Equation 7 produces the same samples as Equation 6.

In our framework, at each time step, we are sampling from a Gumbel-Softmax distribution parameterized by the weights of of the gating module $\Theta_g$. This facilitates the learning of binary decisions in a fully differentiable framework. Following [17], we anneal the temperature from a high value to encourage exploration to a smaller positive value.

# 3 Experiments

## 3.1 Experimental Setup

**Datasets and evaluation metrics.** We adopt two large-scale video classification benchmarks to evaluate the performance of LITEEVAL, *i.e.*, FCVID and ACTIVITYNET. FCVID (Fudan-Columbia Video Dataset) [18] contains $91,223$ videos collected from YouTube belonging to $239$ classes that are selected to cover popular topics in our daily lives like "graduation", "baby shower", "making cookies", *etc*. The average duration of videos in FCVID is $167$ seconds and the dataset is split into a training set with $45,611$ videos and a testing set with $45,612$ videos. While FCVID contains generic video classes, ACTIVITYNET [12] consists of videos that are action/activity-oriented like "drinking beer", "drinking coffee", "fencing", *etc*. There are around $20K$ videos in ACTIVITYNET with an average duration of $117$ seconds, manually annotated into $200$ categories. Here, we use the $v1.3$ split with a training set of $10,024$ videos, a validation set of $4,926$ videos and a testing set of $5,044$ videos. We report performance on the validation set since labels in the testing set are withheld by the authors. For offline prediction, we compute average precision (AP) for each video category and use mean AP across all classes to measure the overall performance following [18, 12]. For online recognition, we compute top-1 accuracy when evaluating the performance of LITEEVAL since average precision is a ranking-based metric based on all testing videos, which is not suitable for online prediction (we do observe similar trends with both metrics). We measure computational cost with giga floating point operations (GFLOPs), which is a hardware independent metric.

**Implementation details.** We extract coarse features with a MobileNetv2 [27] model using spatially downsampled video frames (*i.e.*, $112 \times 112$). The MobileNetv2 is lightweight model and achieves a top-1 accuracy of $52.3\%$ on ImageNet operating on images with a resolution of $112 \times 112$. To extract features from high-resolution images (*i.e.*, $224 \times 224$) as inputs to the fine LSTM, we use a ResNet-101 model and obtain features from its penultimate layer. The ResNet-101 model offers a top-1 accuracy of $77.4\%$ on ImageNet and it is further finetuned on target datasets to give better performance. We implement the framework using PyTorch on one NVIDIA P6000 GPU and adopts Adam [40] as the optimizer with a fixed learning rate of $1e-4$ and set $\lambda$ to 2. For ACTIVITYNET, we train with a batch size of $128$ and the coarse LSTM and the fine LSTM respectively contain $64$ and $512$ hidden units, while for FCVID, there are $512$ and $2,048$ hidden units in the coarse and fine LSTM respectively and the batch size is $256$. The computational cost for MobileNetv2 ($112 \times 112$) ResNet-101 ($224 \times 224$) is 0.08 and 7.82 GFLOPs, respectively.

## 3.2 Main Results

**Offline recognition.** We first report the results of LITEEVAL for offline prediction and compare with the following alternatives: (1) UNIFORM, which computes predictions from 25 uniformly sampled frames and then averages these frame-level results as video-level classification scores; (2) LSTM, which produces predictions with hidden states from the last time step of an LSTM; (3) FRAMEGLIMPSE [41], which employs an agent trained with REINFORCE [29] to select a small number of frames for efficient recognition; (4) FASTFORWARD [4], which at each time step learns how many steps to jump forward by training an agent to select from a predefined action set; (5) LITEEVAL-RL, which is a variant of LITEEVAL using REINFORCE for learning binary decisions. The first two methods are widely used baselines for video recognition, particularly the strong uniform testing strategy which is adopted by almost all CNN-based approaches, while the remaining approaches focus on efficient video understanding.

Table 1 summarizes the results and comparisons. LITEEVAL offers 51.8% (94.3 *vs.* 195.5) and 51.3% (95.1 *vs.* 195.5) computational savings measured by GFLOPs compared to the uniform baseline while achieving similar or better accuracies on FCVID and ACTIVITYNET, respectively. The confirms that LITEEVAL can save computation by computing expensive features as infrequently as possible while operating on economical features by default. The reason that LITEEVAL requires more computation on average on ACTIVITYNET than FCVID is that categories in ACTIVITYNET are action-focused whereas FCVID also contains classes that are relatively static with fewer motion like scenes and objects. Further, compared to FRAMEGLIMPSE and FASTFORWARD that also learn frame usage policies, LITEEVAL achieves significantly better accuracy although it requires more computation. Note that the low computation of FRAMEGLIMPSE and FASTFORWARD results from their access to future frames (*i.e.*, jumping to a future time step), while we simply make decisions whether to

Table 1: **Results of different methods for offline video recognition.** We compare LITEEVAL with alternative methods on FCVID and ACTIVITYNET.

| Method | FCVID | | ACTIVITYNET | |
|---|---|---|---|---|
| | mAP | GFLOPs | mAP | GFLOPs |
| UNIFORM | 80.0% | 195.5 | 70.0% | 195.5 |
| LSTM | 79.8% | 196.0 | 70.8% | 195.8 |
| FRAMEGLIMPSE [41] | 71.2% | 29.9 | 60.2% | 32.9 |
| FASTFORWARD [4] | 67.6% | 66.2 | 54.7% | 17.2 |
| LITEEVAL-RL | 74.2% | 245.9 | 65.2% | 269.3 |
| LITEEVAL | **80.0%** | 94.3 | **72.7%** | 95.1 |

compute fine features for the current frame, making the framework suitable not only for offline prediction but also in online settings, as will be discussed below. In addition, we also compare with LITEEVAL-RL, which instead of using Gumbel-Softmax leverages policy search methods, to learn binary decisions. LITEEVAL is clearly better than LITEEVAL-RL in terms of both accuracy and computational cost, and it is also easier to optimize.

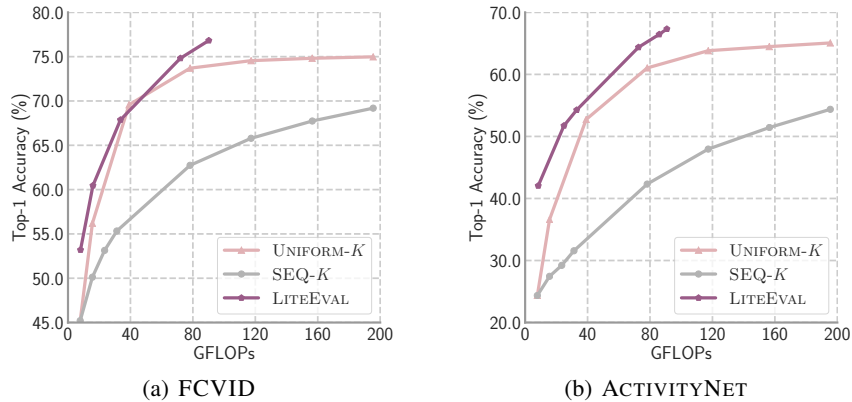

(a) FCVID          (b) ACTIVITYNET

Figure 2: **Computational cost *vs.* recognition accuracy on FCVID and ACTIVITYNET.** Results of LITEEVAL and comparisons with alternative methods for online video prediction.

**Online recognition with varying computational budgets.** Once trained, LITEEVAL can be readily deployed in an online setting where frames arrive sequentially. Since computing fine features is the most expensive operation in the framework, given a video clip (7.82 GFLOPs per frame), we vary the number of times fine features are read in (denoted by $K$) such that different computational budgets can be accommodated, *i.e.* forcing early predictions after the model has computed fine features for the $K$-th time. This is similar in spirit to any time prediction [15] where there is a budget for each testing sample. We then report the average computational cost with respect to the achieved top-1 recognition accuracy on the testing set. We compare with (1) UNIFORM-$K$, which, for a testing video, averages predictions from $K$ frames sampled uniformly from a total of $K'$ frames as its final prediction scores ($K'$ is the location where LITEEVAL produces predictions after having seen the fine features for the $K$-th time); (2) SEQ-$K$, which performs a mean-pooling of $K$ consecutive frames.

The results are summarized in Figure 2. We observe the LITEEVAL offers the best trade-off between computational cost and recognition accuracy in the online setting on both FCVID and ACTIVITYNET. It is worth noting while UNIFORM-$K$ is a powerful baseline, it is not practical in the online setting as there is no prior about how many frames are seen so far and yet to arrive. Further, LITEEVAL outperforms the straightforward frame-by-frame computation strategy SEQ-$K$ by clear margins. This confirms the effectiveness when LITEEVAL is deployed online.

**Learned policies for fine feature usage.** We now analyze the policies learned by the gating module whether to compute fine features or not. Figure 3 visualizes the distribution of fine feature usage for sampled video categories in FCVID. We can see that the number of times fine features are computed

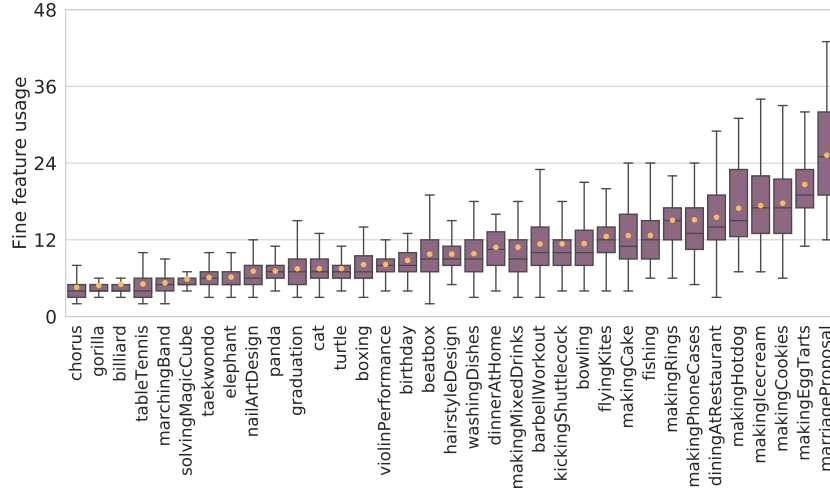

Figure 3: **The distribution of fine feature usage for sampled classes on FCVID.** In addition to quartiles and medians, mean usage, denoted as yellow dots, is also presented.

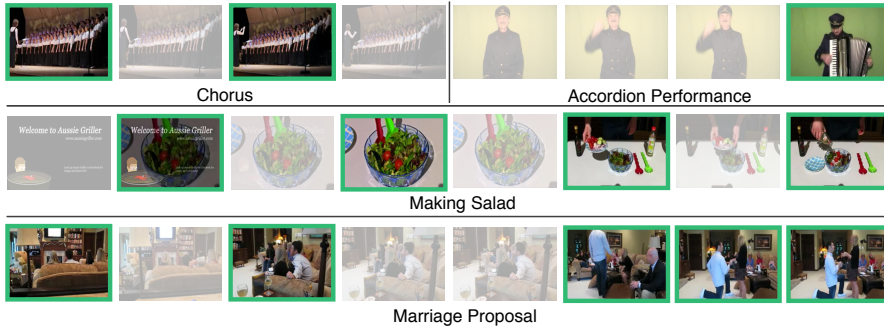

Figure 4: **Frame selected (indicated by green borders) by** LITEEVAL **of sampled videos to compute fine features in FCVID.**

not only varies across different categories but also within the same class. Since fine feature usage is proportional to the overall computation required, this verifies our hypothesis that computation required to make correct predictions is different conditioned on input samples. We further visualize, in Figure 4, selected frames by LITEEVAL to compute fine features of certain videos. We observe that redundant frames without additional information are ignored and those selected frames provide salient information for recognizing the class of interest.

## 3.3 Ablation Studies

**Fine feature usage.** Table 3 presents the results of using $\gamma$ to control fine feature usage in LITEE-VAL. We observe that setting $\gamma$ to 0.05 offers best trade-off between computational cost and accuracies while using a extremely small $\gamma$ (*e.g.*, 0.01) achieves worse results, since it forces the model to compute fine features as seldom as possible to save computation and could possibly overlook important information. It is also worth mentioning that using relatively small values (*i.e.*, less or equal than 0.1) produces decent results, demonstrating there exists a high level of redundancy in video frames.

**The synchronization of the fine LSTM with the coarse LSTM.** We also investigate the effectiveness of synchronization of the two LSTMs. We can see in Table 2 that, without updating the hidden states of the fLSTM with those of the cLSTM, the performance degrades to 65.7%. This confirms that synchronization by transferring information from the cLSTM to fLSTM is critical for good performance as it makes the fine LSTM aware of all useful information seen so far.

Table 2: **The effectiveness of syncing LSTMs on FCVID.**

| Method | mAP |
|---|---|
| w/o. sync | 65.7% |
| LITEEVAL | 80.0% |

Table 3: **Results of different $\gamma$ in LITEEVAL on FCVID.**

| $\gamma$ | mAP | GFLOPs |
|---|---|---|
| 0.01 | 78.8% | 75.4 |
| 0.03 | 79.7% | 82.1 |
| 0.10 | 80.1% | 139.0 |
| 0.05 | 80.0% | 94.3 |

Table 4: **Results of different sizes of LSTMs on FCVID.**

| # units in cLSTM | mAP |
|---|---|
| 64 | 76.9% |
| 128 | 77.3% |
| 256 | 78.3% |
| 512 | 80.0% |

**Number of hidden units in the LSTMs.** We experiment with different number of hidden units in the coarse LSTM and present the results in Table 4. We can see that using a small LSTM with fewer hidden units degrades performance due to limited capacity. As mentioned earlier, the most expensive operation in the framework is to compute CNN features from video frames, while LSTMs are much more computationally efficient—only 0.06% of GFLOPs needed to extract features with a ResNet-101 model. For the fine LSTM, we found that a size of $2,048$ offers the best results.

## 4   Related Work

**Conditional Computation.** Our work relates to conditional computation that aims to achieve decent recognition accuracy while accommodating varying computational budgets. Cascaded classifiers [32] are among the earliest work to save computation by quickly rejecting easy negative windows for fast face detection. Recently, the idea of conditional computation has also been investigated in deep neural networks [30, 15, 24, 6, 1, 22] through learning when to exit CNNs with attached decision branches. Graves [8] add a halting unit to RNNs to associate a ponder cost for computation. Several recent approaches learn to choose which layers in a large network to use [35, 31, 37] or select regions to attend to in images [25, 7], conditioned on inputs, to achieve fast inference. In contrast, we focus on conditional computation in videos, where we learn a fine feature usage strategy to determine whether to use computationally expensive components in a network.

**Efficient Video Analysis.** While there is plethora of work focusing on designing robust models for video classification, limited efforts have been made on efficient video recognition [42, 36, 4, 41, 38, 5, 19, 43]. Yeung *et al.* use an agent trained with policy gradient methods to select informative frames and predict when to stop inference for action detection [41]. Fan *et al.* further introduce a fast forward agent that decides how many frames to jump forward at a certain time step [4]. While they are conceptually similar to our approach, which also aims to skip redundant frames, our framework is fully differentiable, and thus is easier to train than policy search methods [4, 41]. More importantly, without assuming access to future frames, our framework is not only suitable for offline predictions but also can be deployed in an online setting where a stream of video frames arrive sequentially. A few recent approaches explore lightweight 3D CNNs to save computation [5, 43], but they use the same set of parameters for all videos regardless of their complexity. In contrast, LITEEVAL is a general dynamic inference framework for resource-efficient recognition, leveraging LSTMs to aggregate temporal information and making feature usage decisions over time; it is complementary to 3D CNNs, as we can replace the inputs to the fine LSTM with features from 3D CNNs, dynamically determining whether to compute powerful features from incoming video snippets.

## 5   Conclusion

We presented LITEEVAL, a simple yet effective framework for resource-efficient video prediction in both online and offline settings. LITEEVAL is a coarse-to-fine framework that contains a coarse LSTM and a fine LSTM organized hierarchically, as well as a gating module. In particular, LITEEVAL operates on compact features computed at a coarse scale and dynamically decides whether to compute more powerful features for incoming video frames to obtain more details with a gating module. The two LSTMs are further synchronized such that the fine LSTM always contains all information seen so far that can be readily used for predictions. Extensive experiments are conducted on FCVID and ACTIVITYNET and the results demonstrate the effectiveness of the proposed approach.

**Acknowledgment** ZW and LSD are supported by Facebook and the Office of Naval Research under Grant N000141612713.

## Footnotes

* Part of the work is done when the author was an intern at Salesforce Research.

[1] We absorb the weights of the classifier $\boldsymbol{W}_p$ into $\Theta_{\texttt{fLSTM}}$.

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
