[Reviews · NeurIPS 2019]

Reviewer 1



this paper is well-written and intuitive. the approach sounds technically correct. the novelty is moderate. The paper doesn't aim for a significant technical contribution. Instead, it tackles the valid problem of video recognition using a limited computational budget. the related work is short and some works are not covered: -SlowFast Networks for Video Recognition -Skip-RNNs in terms of results, it shows the benefit of conditional gating for processing fine features. the experiments are relatively convincing but its improvement is not significant I believe. - in Table 1, the results for the UNIFORM method is shown for 25 uniformly sampled frames, which results in almost 2x GFLOPs compared to the proposed method. However, Fig.2 shows that to obtain the same accuracy, uniform sampling needs slightly more frames than the proposed method (~1.3x).

Reviewer 2



This paper focuses on the video classification task, its goal is to speed up the inference of typically heavy video neural networks. The proposed method is a "cascade" like approach where one light-weight 2D CNN (e.g. MobileNets) scans through all frames, and an LSTM is employed to decide whether to apply a heavier-weight 2D CNN (e.g. ResNets) on each frame. The main innovation of this paper, is that opposed to previous work which are reinforcement learning-based, it proposes to use the Gumbel softmax trick which allows the whole framework to be trained end-to-end. Empirical results on FCVID and ACTIVITYNET confirms that such optimization process outperforms several RL-based baselines. Clarity: the paper is very clearly written and easy to read. Originality: Although the proposed framework is not entirely novel (the high-level idea of skipping frames, and cascades have been explored by previous work on this topic), the optimization process is novel for this particular application (although explored under other scenarios) and shown to be critical by empirical evaluations. Significance: speeding up video classification networks is important for practical applications. However, the choice of using LSTM to aggregate temporal features may limit its practical application, as it has been shown by recent work that 3D ConvNets significantly outperform RNN alternatives on video classification benchmarks. This can also be seen from Table 1, where the uniform baseline is "unreasonably" strong compared with LSTM, on both datasets. (Post rebuttal) The reviewer appreciates the authors' rebuttal which addressed my concerns. I keep my original rating and recommend acceptance of the paper.

Reviewer 3



Originality: The task is not new, as the authors mentioned in the related work section, many previous literature have worked on this problem. The proposed method is very similar to this paper (Low-Latency Video Semantic Segmentation, CVPR 2018, and other literature as well). Hence the novelty of the paper is limited. Quality: I have one concern about the gating mechanism. How to justify it is really working? The gating mechanism is a one layer MLP, outputting the probability of whether to extract fine details of the video frames or not. Maybe it is just doing a random selection. The performance gap between the proposed method and the uniform baseline is really small, which is hard to justify its effectiveness. Another thing is, for video classification, usually we pick Kinetics, UCF101, something-something as the evaluation dataset. This submission choose FCVID and ActivityNet, which is hard to compare to previous approaches. And again, it is hard to justify the method's effectiveness. Clarity: This submission is well written and easy to understand.

[Author Response · NeurIPS 2019]

We thank the reviewers for their constructive feedback.

**The use of LSTMs instead of 3D CNNs [R1, R2, R3].** We use LSTMs as an "agent" not only to make classification
predictions but more importantly to make *sequential gating decisions—dynamically* determine whether to compute
features at a finer scale conditioned on incoming video frames and historical information. In particular, the decision for
the $t$-th time step depends on previous observations and decisions—if the current frame is very similar to previous seen
frames (this is very common as there are a lot of redundant frames) the model is more likely to use coarse features as no
new information is provided. Therefore, we use LSTMs to learn the fine feature usage policy based on the history of
past interactions with the video, which is similar in spirit to a MDP process if we replace Gumbel-Softmax with RL. In
addition, the autoregressive nature of LSTMs allows LITEEVAL to save computation for online video recognition with
minimal modification, while it is not clear how to use uniform sampling for online recognition since it is hard to select
the optimal sampling rate for different videos as there is no information (*i.e.*, duration and fps) known beforehand about
incoming videos (sampling every 1 min might work for long videos but would be problematic for a 1-min-long video).

Existing 3D CNN models (I3D, S3D, *etc*) typically average prediction scores from $N$ (*e.g.*, 10/25) uniformly sampled
snippets (stacked frames) as video-level predictions. There are a few disadvantages: (1) they operate on snippets,
requiring storing multiple frames for 3D convs, which is not feasible on low-power devices; (2) 3D CNNs are generally
computationally expensive (108 GFLOPs for a single snippet in I3D); (3) they produce one-size-fits-all models for all
videos regardless of their complexity; (4) the uniform sampling strategy for testing prevents them to be readily used in
online settings. In contrast, our model saves computation *for both online and offline settings* by using expensive features
as infrequently as possible. Note that we could use 3D CNNs as our feature extractor when computational budget is
sufficient. To study whether our framework is compatible with modern frameworks for video recognition, we adopted a
DPN model trained using the temporal segment network, which is a state-of-the-art framework for video recognition;
LITEEVAL offers 83.6% on ACTIVITYNET, confirming LITEEVAL supports features from different backbones.

**Comparisons with SlowFast [R1, R2, R3].** Thanks for pointing out this paper, which will be cited and discussed. But
we do like to stress that there are significant differences between our approach and SlowFast—(1) SlowFast produces
the same set of parameters for all videos whereas our approach allocates computational resources conditioned on input
videos; (2) SlowFast relies on the uniform sampling baseline, making it unsuitable for online recognition; (3) SlowFast
operates on video frames with the same spatial resolution (*i.e.*, $224 \times 224$) and uses lightweight CNNs for the Fast
pathway ($\sim 20\%$ computation) and heavy CNNs for the Slow pathway. In our model, we not only use a lightweight
CNN to extract coarse features but also reduce the input resolution, making the computation overhead of the coarse
features negligible (0.08 GLOPs). We are currently preparing a comparison.

**Combining coarse and fine features [R1].** As suggested by the reviewer, we also compare with the uniform sampling
and the LSTM baseline using both coarse and fine features on ACTIVITYNET. Although fusing two features does
slightly improves the performance of using fine features alone, we observe that LITEEVAL still achieves better results
compared to the uniform sampling (72.7% *vs.* 70.6%) and the LSTM baseline (72.7% *vs.* 71.5%).

**Second term in the loss/syncing cLSTM and fLSTM [R1].** The ablation study of synchronizing the LSTMs are
reported in Tab. 2. The 2nd term in the loss function controls the computational budget and results are shown in Tab. 3.

**Comparison with Skip-RNNs [R1].** Skip-RNNs achieved similar mAP as our LSTM baseline with slightly less
computation, since it processes *every frame* and saves computation by learning to skip updates of RNN models. Note
that the most expensive computation in a video recognition pipeline is feature extraction (7.82 GFLOPs for a single
frame with a ResNet101), and the computation incurred by LSTMs is negligible (0.005 GFLOPs per time step).

**Numbers in Tab 1 and Fig. 2 [R1].** Tab. 1 reports the performance of LITEEVAL in an offline setting while Fig. 2
summarizes online recognition results. Please refer to L221-L225 for more details.

**Gating [R3].** We vary the computational budget of LITEEVAL by adjusting $\gamma$ and then compare with random selection
that uses similar computation budget during inference on ACTIVITYNET. The mAP (random *vs.* LITEEVAL) is 65.8%
*vs.* 72.7% (102 GFLOPs); 69.1% *vs.* 73.2% (183 GFLOPs). In addition to qualitatively visualized selected frames
in Fig. 4, we also visualized *quantitatively* fine feature usage statistics of selected classes in Fig. 3, we can see that
for simple classes like objects (*e.g.*, gorilla), LITEEVAL makes predictions with less fine feature usage, while more
computation is needed for more complicated classes (*e.g.*, marriage proposal). This confirms LITEEVAL is able to learn
useful gating decisions.

**Datasets [R3].** We chose FCVID and ACTIVITYNET for evaluation since videos in these two datasets are "untrimmed"
with an average duration over 100 seconds whereas videos in UCF-101 ($\sim$8s), Kinectices ($\sim$10s) and Something-
Something ($\sim$4s) are all "trimmed". Compared to extensive research efforts on action recognition on trimmed videos,
we believe long-term video understanding is also a very important and arguably a more challenging problem; resource-
efficient models for long videos are of great value for their deployment in real-world scenarios.

[Meta-Review · NeurIPS 2019]

This paper presents research on resource efficient video analysis. The reviewers appreciate the frame gating approach and solid methodology to perform a dynamic decision on inference-time resources that should be used for classifying an input video. The model is fully differentiable (via Gumbel-softmax), in contrast to RL-based approached for learning similar frame-skipping methods. The reviewers also note that the empirical evaluation is solid, with good comparisons to baselines/ablation studies. While there are some concerns regarding the magnitude of the contributions (e.g. relevance with LSTM-based models vs. other temporal deep learning architectures) and novelty, on balance it is a solid, well-written paper that makes a clear contribution to efficient video analysis.